# Comparative Analysis of Resident Space Object (RSO) Detection Methods

**DOI:** 10.3390/s23249668

**Published:** 2023-12-07

**Authors:** Vithurshan Suthakar, Aiden Alexander Sanvido, Randa Qashoa, Regina S. K. Lee

**Affiliations:** 1Department of Earth and Space Science, York University, Toronto, ON M3J 1P3, Canada; randaq@my.yorku.ca (R.Q.); reginal@yorku.ca (R.S.K.L.); 2Department of Electrical and Computer Engineering, McMaster University, Hamilton, ON L8S 4K1, Canada; sanvida@mcmaster.ca

**Keywords:** space situational awareness (SSA), resident space objects (RSOs), detection algorithm, optical images

## Abstract

In recent years, there has been a significant increase in satellite launches, resulting in a proliferation of satellites in our near-Earth space environment. This surge has led to a multitude of resident space objects (RSOs). Thus, detecting RSOs is a crucial element of monitoring these objects and plays an important role in preventing collisions between them. Optical images captured from spacecraft and with ground-based telescopes provide valuable information for RSO detection and identification, thereby enhancing space situational awareness (SSA). However, datasets are not publicly available due to their sensitive nature. This scarcity of data has hindered the development of detection algorithms. In this paper, we present annotated RSO images, which constitute an internally curated dataset obtained from a low-resolution wide-field-of-view imager on a stratospheric balloon. In addition, we examine several frame differencing techniques, namely, adjacent frame differencing, median frame differencing, proximity filtering and tracking, and a streak detection method. These algorithms were applied to annotated images to detect RSOs. The proposed algorithms achieved a competitive degree of success with precision scores of 73%, 95%, 95%, and 100% and F1 scores of 68%, 77%, 82%, and 79%.

## 1. Introduction

Space situational awareness (SSA) has been a growing concern around the world, as space is being reevaluated as a domain that needs protection to sustain a nation’s sovereignty and monitor its assets in space. The number of resident space objects (RSOs) has remarkably increased in recent years. This occurrence is related to the Kessler Syndrome, which describes the probability of a collision between RSOs cascading into further collisions. Due to the substantial surge in satellite launches and trends towards mega-constellations, there are now numerous manufactured elements orbiting in near-Earth space. Among these objects are fragments of detonated bolts from separated upper stages of rockets, as well as entire satellites without an end-of-life plan.

In the past, the space industry operated on the “Big Sky” mentality, which presupposes that the volume of space—compared to debris fragments—is so vast that it is unlikely two objects will ever collide. However, the Department of Defense’s global Space Surveillance Network (SSN) sensors currently track 34,810 debris fragments [1]. Furthermore, the European Space Agency (ESA) estimated that there were 640 past collisions and fragmentation events [1]. Therefore, it is crucial to further develop SSA to prevent collisions involving hypervelocity objects such as rocket bodies, which can lead to debris clouds that endanger the near-Earth ecosystem.

With the sudden and rapid increase in RSOs, there has also been a substantial push to advance SSA technologies in recent years. Much effort has been made for the development and demonstration of sensor technologies ranging from optical imagers with various wavelengths to technologies such as light detection and ranging (LiDAR), lasers and other sensing techniques.

### 1.1. Overview of RSO Imaging Technologies

#### 1.1.1. LiDAR and RADAR Systems

LiDAR, a remote sensing technology that employs lasers to measure distances and create detailed three-dimensional maps of objects and environments, has been crucial in tracking and monitoring RSOs. These data can be used for collision threat assessment or to provide detailed information for mitigation plans. In [2], the researchers suggested an SSA mission that utilized LiDAR technology for a low-Earth-orbit spacecraft. Additionally, References [2,3,4] are relevant studies that explored LiDAR in SSA. Radio detection and ranging (RADAR) has also been considered for SSA applications [5,6,7,8]. Radio waves are emitted to estimate the distance, speed, and trajectory of RSOs in a similar fashion to that used with LiDAR sensors. Space-borne radar has been used to track both active satellites and inactive RSOs in a way that resembles the approach used in LiDAR systems. In [9], RADAR systems (space-borne sensors) were categorized as “active” sensors that emit energy and measure the returned signal to calculate the difference between the emitted and the returned signal. In contrast, optical sensors (including infrared and visible-range sensors) were considered “passive” sensors.

#### 1.1.2. Optical Systems

Our efforts in SSA imaging primarily focus on optical imaging using both space-based and ground-based imagers—specifically, optical telescopes. In the active tracking mode, an imager moves to track an object of interest. In contrast, the passive mode involves the camera imaging a star field in its field of view without any motion being imposed on the imager itself. Traditionally, star trackers installed on satellites have imaged star fields within their fields of view in a manner similar to this imaging process to ascertain their altitude. These telescopes, whether in space or on the ground, are the primary tools for tracking and observing objects in space. They collect photometric and astrometric data by capturing reflected light from RSOs, thus aiding in object identification and characterization. Space-based optical sensors offer distinct advantages, such as uninterrupted, high-quality observations that are unaffected by Earth’s atmosphere. When positioned in low Earth orbit, they provide global coverage with predictable revisit times and are immune to weather and lighting conditions, including day/night cycles and light pollution. Recent SSA missions using optical telescopes included the Cosmos series (Russia), Tiangong Space Station (China), the Sentinel series (European Union), and the BRITE constellation [10]. Notable Canadian SSA missions, such as Sapphire and NEOSSat, which were launched in 2013, relied on passive optical payloads for imaging, detecting, and monitoring RSOs in Earth orbits. Another microsatellite space surveillance mission is currently under development and will continue using this optical approach.

Space-borne optical sensors have a critical drawback compared to their ground-based counterparts, namely, their limited field of view (FOV), making continuous monitoring of a satellite’s surroundings challenging. While the costs of sensors are similar, the overall cost of satellites, operations, and launches far exceeds that of ground-based observations. These sensors are placed on satellites with strict constraints on their mass, data, power, volume, and scheduling, which complicates the achievement of continuous coverage. In contrast, ground-based optical sensors offer a low-cost, extensive, and flexible observation solution compared to space-borne sensors. However, they are limited to nighttime operation, and their effectiveness is affected by atmospheric turbulence, geographic location, and weather conditions, making them less suitable for advanced SSA missions.

### 1.2. Research Objectives

Once an RSO is imaged, regardless of the wavelength or FOV of the imager, the next step in the imaging process is to detect, identify, and characterize the RSOs. Depending on the mode of observation (passive mode vs. active tracking mode), an RSO may appear as a moving object—a dot across the FOV or a streak. Object detection and streak detection are related concepts in the context of RSO observation and imaging, but they refer to slightly distinct aspects of detecting objects in motion. Object detection involves identifying and locating objects within an observed scene. In the context of RSO observation, this refers to identifying moving objects against the background of the dark sky with stars (often seen as stationary objects relative to the RSO’s movement). Streak detection, on the other hand, refers to the process of identifying streak-like patterns in images that are caused by fast-moving objects. In RSO observation, these streaks are usually caused by objects in orbit passing through the field of view of a sensor during the time in which the image is being captured with a relatively long exposure time. Streak detection involves recognizing these trails in images and associating them with RSOs, such as satellites or space debris. Streak detection is commonly used in space surveillance to identify objects that might otherwise be too faint or fast-moving to be easily discerned. In this study, we use both object and streak detection with images taken with a short exposure time. Streaks are created by stacking multiple images instead of using long-exposure observations.

Note that object detection and object tracking are two distinct processes that involve observing and monitoring objects in space in the context of space situational awareness (SSA). In the current study, we focus on object detection only to identify an RSO present within optical images. Once an RSO is detected, its position and attitude within the observed scene are determined with its photometric and/or astrometric information. Based on these data, the RSO’s characteristics, such as its orbit, size, shape, optical properties, and potentially its identity, are estimated. This last step is often referred to as identification and characterization. In this process, object detection is crucial for identifying new objects with the accurate and precise location of their centroids within the frame. Object tracking, on the other hand, is the process of continuously monitoring the movement and trajectory of an RSO over time. Once an RSO has been detected and located within the FOV, the sensor is programmed to track its trajectory based on the predicted patterns of movement. Particularly, in both ground- and space-based photometric observation of an RSO for the estimation of the attitude and optical properties according to a light curve analysis, tracking is a critical step. In this paper, we focus on passive-mode observation using wide-FOV imagers with a reasonably short exposure time to yield images with moving objects. Tracking was not considered for the purposes of the current study.

The images analyzed in this study were captured as “observations of opportunity”, in which the imager was not tasked with observing planned targets. Similar examples include the study of RSO image analysis using the Fast Auroral Imager (FAI) described in [11] and the operation of dual-purpose star trackers discussed in [12,13,14]. Further details on the mode of operation in similar scenarios can be also found [15]. In Section 2, we present a survey on the object detection algorithms used in SSA applications. Section 3 provides a detailed description of the detection algorithms that we tested and implemented in the current study, namely, adjacent frame differencing (AFD), median frame differencing (MFD), proximity filtering and tracking (PFT), and streak detection. Before the results are described in Section 5, we outline the datasets used in this study in Section 4. The conclusion and future work are discussed in Section 6.

## 2. Survey of Object Detection Algorithms in SSA Applications

Object detection methods are continuously evolving, with numerous algorithms emerging annually, making it challenging to compile an all-encompassing survey. Instead, several survey articles have provided insights into this rapidly changing field. Ragland and Tharcis [16] focused on traditional (non-AI-based) methods, including frame differencing, optical flow, and background subtraction. Meanwhile, the authors of [17] offered a comprehensive review of deep learning techniques for object detection, and they addressed datasets, assessment metrics, context modeling, and detection proposal methods. The evolution of object detection elements, from detectors to optimization techniques, from 1998 to 2021 was covered in [18]. A holistic view of deep learning systems, methods, benchmark datasets, and real-world applications can be found in [19].

Beyond detection, object tracking and streak detection are both widely studied fields in vision science, with a wide range of applications in autonomous vehicles, transportation management, and health monitoring. Several techniques have been developed and applied to these fields with varying levels of success. As AI-based techniques have been successfully implemented, they have led to the emergence of many algorithms, which are supported by readily available datasets for training and validation. However, these topics are outside the scope of the current study and will be considered later.

The application of most of the methods described above to RSO detection has not been extensively studied, and no conclusive results are available. A recent paper by Massimi et al. offered the most comprehensive survey on the subject [20]. The authors classified methods into non-AI- and AI-based methods. Among the AI-based methods, YOLO, CNN, and other branches of deep learning were also described. Additionally, they presented preliminary results from a case study on SSA applications. The case study presented in the referenced literature primarily focused on simulated environments and radar processing. While AI-based methods, which are primarily deep-learning approaches, do show promising results, a significant technological gap persists in the implementation of these methods in real-time scenarios. Implementing AI-based object detection methods can be a complicated task that requires the integration of various technologies and for several challenges to be addressed. The primary challenge in an AI application is the collection, annotation, and validation of data to ensure their quality and quantity. Deep learning models necessitate significant volumes of high-quality labeled data for training. Acquiring, curating, and annotating such data can be expensive and time-consuming. Ensuring the precision and diversity of the training dataset is fundamental for a model’s performance. In SSA applications, there are no publicly available annotated datasets for evaluating newly developed algorithms.

For RSO images, we examined various datasets composed of ground- or space-borne images of star fields that contained RSOs, such as NEOSSat images [21], FAI images from onboard CASSIOPE [22], and ground observations from various telescopes and observatories around the world [23]. However, creating accurate annotations for object detection tasks is time-consuming, challenging (given the complex nature of the images), and expensive (as it requires trained personnel who can recognize a moving object as an RSO in star field images). Secondly, defining bounding boxes or segmentation masks for objects in images also requires expertise and meticulous attention to detail. Errors in annotations can lead to biased or incorrect model predictions. Third, choosing the right object detection architecture and configuration is also difficult, and the performance relies heavily on the nature of the images with which the algorithms are trained. Different models have varying trade-offs between accuracy and speed. Tuning hyperparameters and architecture choices requires extensive experiments, testing, and experience. Lastly, most AI-based methods require extensive computational resources and powerful hardware such as GPUs or TPUs for training and inference. Implementing and scaling such models can be costly and can require careful resource management, making it nearly impossible to implement them onboard spacecraft in real time.

Given these difficulties with AI-based methods, the current study is intended to gain an understanding of the applicability of the popular non-AI-based detection methods specifically in SSA applications and aims to provide a baseline for future development. We also provide an annotated dataset that was collected and pre-processed from a stratospheric balloon-borne platform on which the current study was based. The accompanying dataset can be used as a training and validation set to develop AI-based methods and to improve RSO detection algorithms.

## 3. RSO Detection Methods

Detecting RSOs in star field images is a challenging task for various reasons. RSOs are smaller and less intense than stars, which largely dominate star field imaging. Consequently, RSOs appear relatively faint compared to stars, which are luminous objects. The brightness of a Resident Space Object (RSO) stems from the phase angle, which is the angle between the Sun and the observer, as measured at the RSO itself. However, the brightness of RSOs is not consistent during an observed transit. Significant differences in brightness are evident between various RSOs, such as active satellites, tumbling satellites, and rocket bodies [24]. In addition, the inherent image noise due to elements such as sensor noise and cosmic rays can also suppress the photometric value of RSOs, which further complicates detection. Especially for images acquired using ground-based systems, additional noise, such as light pollution, atmospheric effects, and changing lighting conditions, presents hindrances to the task of RSO detection due to the varying background. Furthermore, most RSOs lack distinctive features and are subject to environmental degradation in space, making it hard to differentiate between them and stars. The spatial resolution of the imaging system can heavily influence the distinction process, although higher-resolution images are scarce. RSOs travel at varying velocities, thereby making it difficult to identify multiple RSOs that are present in a singular image sequence.

Hence, all factors must be considered in RSO detection amidst star fields to differentiate RSOs from the background stars. The detection can be made easier with preprocessing that accounts for the characteristics of the imaging system, intrinsic and extrinsic calibration, and an understanding of the orbital parameters of RSOs that are likely to be observed. While a multitude of object detection methods are reliable, efficient, and effective methods in other applications, not all of them would be sufficient for RSO images. We acknowledge that there is a plethora of object detection and tracking methods. It is to be noted that we intentionally excluded AI-based approaches from this comparison, as our aim was to explore detection capabilities through implicit methods, which are deemed more appropriate for our dataset. An AI-based approach for RSO detection will be explored with this dataset in future research.

In this study, we largely evaluated four algorithms for comparison purposes. We chose three frame differencing techniques, namely, adjacent frame differencing (AFD), median frame differencing (MFD), and proximity filtering and tracking (PFT), for object detection. We also briefly considered the optical flow and nearest neighbor methods, but based on the literature review, they were insufficient for RSO detection. Instead, we implemented streak detection with the aid of plate solving. A comparison among these detection methods is summarized in Table 1. The detection methods are fully automatic and designed for autonomous real-time processing onboard satellites. The only manual intervention occurred during the initial annotation phase of the dataset, where experts performed hand-labeling to create accurate validation data. The frame differencing methods proposed in this study have been extensively researched in various areas of computer vision. These methods are predominantly used for motion detection in videos from surveillance cameras (also known as CCTV (closed-circuit television)). Adjacent frame differencing is employed in enclosed environments, such as shopping complexes and theaters, where a comparison between current and previous images is utilized to isolate objects of interest. When monitoring traffic on highways, median frame differencing has been used to detect objects in motion by subtracting a background frame from the current frame. However, further exploration is necessary for their application in space situational awareness (SSA). Hence, this study aims to apply these well-established methods to a novel SSA dataset. The results of this study will be used for implementation onboard a satellite for real-time resident space object (RSO) detection in the long term. The non-AI-based methods outlined in [20] have not been evaluated with space situational awareness (SSA) images. Given the current limitations, a direct comparison among the existing algorithms is not feasible at this time. Future research could focus on a more direct comparison of the proposed algorithms while using similar datasets to further validate the findings of our study.

### 3.1. Adjacent Frame Differencing (AFD)

The AFD algorithm relies on subtracting adjacent frames within an image sequence to isolate moving objects. During the preprocessing stage, each frame in the sequence is first normalized to reduce the range of pixel intensity values. Normalization is necessary for the OpenCV software library to further modify the sequence, as images must be converted to an 8-bit format before processing occurs. After normalization, the algorithm iterates through the sequence and calculates the absolute difference in pixel intensities between the current and subsequent frames. The resulting differences in intensities are used to generate a new frame, with white pixels representing motion. Overlapping stars between adjacent frames result in black pixels, as subtracting identical pixel intensities results in a difference of zero. Following subtraction, size filtering is applied to remove remaining hot pixels and visual artifacts with an area smaller than a predefined threshold. The resulting frame is processed using OpenCV’s contour detection function, and groupings of white pixels are identified as RSOs. Bounding boxes are then fitted to each contour to encapsulate the detected RSOs. Finally, the center coordinates of each bounding box are tabulated. A flowchart of the detection process with the adjacent frame differencing method is presented in Figure 1. Various frames are visualized in Figure 2. In the presence of a significant frame time, RSOs do not overlap between adjacent frames and, thus, are detected twice after subtraction. To mitigate double detections, overlapping bounding boxes in adjacent frames are combined, resulting in a single detection.

### 3.2. Median Frame Differencing (MFD)

MFD involves subtracting each frame in an image sequence from a median frame in order to isolate moving objects. During preprocessing, each frame in the sequence is normalized and thresholded, similarly to the process in AFD. The operational flow diagram for the median frame differencing (MFD) algorithm is shown in Figure 3. The input sequence is then divided into equal segments of an arbitrary length, with a unique median frame being generated for each segment. Median frames are constructed by averaging the intensity of pixels in each frame throughout a segment. Frames within a segment are then subtracted from their corresponding median frames, revealing moving objects as white pixels, as illustrated in Figure 4. Visual artifacts and hot pixels are eliminated using size filtering, and contours with a greater concentration of black pixels near their center are discarded. This filtering technique is effective in MFD as it does not capture moving objects in the median frame. As a result, RSOs retain their shape after subtraction due to the lack of overlap between the current frame and the median frame. Thus, stars are distinguished from RSOs by white pixels surrounding a dark centroid, while RSOs appear as tight groupings of white pixels. By leveraging this visual distinction, MFD accurately differentiates stars from RSOs. The resulting frame is then processed using OpenCV’s contour detection, which identifies groupings of white pixels as RSOs. As in AFD, bounding boxes are subsequently fitted to each contour, and the center coordinates of each bounding box are then recorded.

### 3.3. Proximity Filtering and Tracking (PFT)

PFT operates by generating a median frame, detecting all contours present, and fitting bounding boxes to each contour. The center coordinates of each bounding box are then stored in a list for later use. Figure 5 provides a functional flow diagram of PFT. The algorithm subsequently scans through each frame in the sequence to identify objects using OpenCV’s contour detection, as presented in Figure 6. Any objects observed to be in close proximity to the previously saved coordinates are labeled as stars and promptly discarded. The remaining objects are tracked using a unique numerical ID that persists throughout the sequence. Also, moving objects that travel an insignificant distance between their initial and final appearances in the sequence are discarded. This prevents the detection of hot pixels, which appear in a single frame. Finally, bounding boxes are fitted to each remaining object, and the center coordinates of each bounding box are stored.

### 3.4. Streak Detection from Stacked Short-Exposure RSO Images

As an alternative to the frame differencing techniques described above, we deployed streak detection utilizing plate solving to prevent additional blurring from stacked images. As previously discussed, a series of short-exposure images can be stacked to create an image with RSOs in a line, resembling a streak.

One commonly used technique for streak detection is achieved with the use of edge detection, an image processing technique, which identifies abrupt changes in image intensity—typically at region boundaries—and defines these changes as edges. The Canny edge method stands out as a preferred choice due to its superior performance [25]. The Canny method involves several steps: An initial Gaussian filter is applied to reduce image noise, then the image gradient is evaluated using amplitude and angle calculations, thus refining the outlines. The final step includes a hysteresis-based thresholding process to preserve the desired edges while removing undesired ones. Subsequently, the Hough transform complements Canny edge detection by verifying the linearity of the edge and accurately estimating streaks. However, despite the denoising in the Canny method, the Hough transform may generate multiple lines for the identification of a single streak. Consequently, clustering is employed to isolate the specific streak, though this process can be time-consuming and susceptible to image noise.

In the current study, a stacking process was used to generate streaks from a set of twenty-seven individual images (more details on the data acquisition will be provided in Section 4). In the resulting stacked images, the stars appeared stationary, while RSOs were rendered as streaks. The stacking method provides numerous benefits, including decreased processing time, less of a need for computational resources, and an enhanced signal-to-background ratio for streak detection. However, the quality of the stacked images can be significantly impacted by the observer’s motion, with the balloon’s stability having a direct impact on the outcome.

Identifying “streaks” from star field images requires a step to remove stars. The stacked images underwent plate solving via Astrometry.net. This was chosen for its robustness, as it eliminated the need for additional information regarding calibration. Astrometry.net employs various image processing techniques to precisely detect hundreds of stars, achieving sub-pixel accuracy. Each quad, which consists of four stars, is analyzed to assign a geometric hash based on the relative positions [26]. A Bayesian decision problem is employed to validate this alignment assessment, resulting in rare occurrences of false-positive matches [26]. For wide-angle images, the Tycho-2 catalog was utilized [26]. Once processed, the output files contained World Coordinate System (WCS) transformations, which provided correspondences between the pixel locations in the images and reference stars.

The WCS information was used to ascertain the pixel positions of all identified stars. For each of these located stars, a circular mask was created by calculating the squared distance between each detected star pixel in the image and the circle’s center by utilizing the square of the circle’s radius. Pixels residing in the inner regions of masks were then set to zero, effectively mitigating false positives caused by the presence of stars within these regions. Figure 7 provides a visual representation of an image before and after the removal of stars, with the scales indicating pixel intensity. The left side depicts the original image, while the right side shows the same image with the stars removed. The names of the stars and their BT and VT magnitudes (blue and visual magnitudes) were identified using the right ascension (RA) and declination (Dec) values from the header information. Additionally, the Euclidean distance between the stars’ positions in the images and those in the index was calculated.

Dynamic thresholding was used to set regions with pixel intensities below the background brightness values to zero. Subsequently, the images were binarized to enable the use of OpenCV’s "find contour" function. To minimize false detection, the area of the detected contours was evaluated, and an area-based threshold was applied. The area of the contours needed to exceed that of the largest contour in the image minus three times the standard deviation of the detected contours. This allowed for the removal of image artifacts. For each remaining contour, a mask was formed to isolate pixels situated above the background within the contour. The mask’s minimum and maximum values identified the pixel coordinate of the streaks’ endpoint. The distance between these coordinates was calculated to determine the streak’s length. The streak length needed to exceed the radius used during star removal. The signal-to-background ratio (SBR) of the streaks was evaluated using the formula mentioned in [27]. A functional flow diagram of the algorithm is presented in Figure 8.

## 4. Dataset Used in the Current Study

The scarcity of labeled data presents the greatest challenge in many applications in which true data are needed for training and verification. RSO images encounter a similar issue due to the absence of publicly accessible labeled data for algorithm design. Due to the shortage of labeled data, the researchers opted to use a simulated dataset containing synthetic streaks and RSOs. The simulated dataset primarily utilized BUAA-SID 1.0 and SPARK [28,29]. BUAA-SID 1.0 comprises over 4000 images that were created by uniformly sampling around a satellite model with a single external light source. This dataset includes 20 distinct satellite models, featuring a variety of types, shapes, and functionalities, and they were all imported into the 3dsMax software. However, it lacks a simulation of the space environment. The SPARK dataset employs real models of 10 different satellites and spacecraft, in addition to five types of space debris. Nonetheless, it does not feature a comprehensive range of space debris models. Several other simulated SSA datasets focusing specifically on the estimation of the poses of spacecraft have been developed. Nevertheless, these datasets lack several elements found in the actual space environment. The only publicly accessible event-based space imaging dataset is provided by Western Sydney University [30]. Nonetheless, their sensors were ground-based telescopes, and there are inherent limitations of ground-based observations and challenges associated with the use of event-based sensors.

Numerous studies have successfully demonstrated the accuracy and precision of RSO detection in simulated environments [31,32,33]. In contrast, implementing the proposed methods when using real-world images presents several challenges, including the presence of dead pixels, hot pixels, and cosmic rays, the unknown point spread functions of imagers, various effects that radiation imposes on the imager, and substantial stray light from sources such as Earth’s albedo, the Moon, and the Sun. In particular, with space-borne images such as FAI, we have observed numerous scenarios in which the background lighting conditions (e.g., the south anomaly) make RSO detection extremely difficult. Therefore, for the purposes of this study, we tested the RSO detection algorithms with the dataset that we collected from the stratosphere and manually annotated for accuracy.

As noted earlier, manual labeling is considerably challenging due to the nature of these images. Annotating medical images, for example, is equally challenging and requires skilled experts. Similarly, the Cityscapes dataset explores the intricacies of real-world urban scenes and the complexity behind annotating them for applications in autonomous vehicles [34]. All of these challenges exist with RSO images. Distinguishing RSOs (moving dots) from stars (non-moving dots) in monochromatic images proves to be extremely challenging when relying solely on the naked eye. Additionally, the limited stability of an imaging platform that is also in motion presents an additional difficulty in annotation. The difficulties with RSO detection outlined in Section 3 were also hurdles that the annotators had to overcome.

### RSONAR Mission Overview

The SSA dataset utilized in this paper was derived from the Resident Space Object Near-Space Astrometric Research (RSONAR) mission, which was launched in August 2022 on a stratospheric balloon platform. The payload of RSONAR provided a distinctive vantage point in the stratosphere to effectively observe RSOs, especially during dawn and dusk, when the phase angles were favorable for RSO observation. The of payload RSONAR featured the pco.panda 4.2 from PCO Imaging based in Kelheim, Germany, which is a scientific complementary metal-oxide semiconductor (sCMOS) camera [35], complemented by a Zeiss 2229-998 C-Mount Industrial Lens obtained from Carl Zeiss Industrielle Messtechnik GmbH in Oberkochen, Germany [36]. The key specifications for the PCO camera used during the flight are detailed in Table 2. Comprehensive details regarding the hardware, operations, and payload development can be found in [37]. The mission, which was conducted as part of the Canadian Space Agency’s STRATOS program in collaboration with the Centre National d’Études Spatiales [38], saw the payload operate for nearly 9 of the 13.73-hour day–night flights, from 3:27 to 12:24 UTC, while utilizing a strategic ballast release and pointing adjustments to maintain stability at altitudes between 30 and 45 km amidst minimal atmospheric turbulence.

The main payload of RSONAR (with a PCO sCMOS camera) obtained 95,046 images through passive-mode observations. The dataset encompassed various elements, including star streaking as the balloon ascended, ballast deployment, and the intrusion of sunlight into the sensor at the end of the flight. The stability of the gondola played a critical role in minimizing the camera jitter due to the nature of the platform. To ensure continuous data acquisition throughout the operational period, twenty-seven images were collected in bursts, with a 4 s delay between each burst.

For this study, 429 images from this dataset were annotated. The images were categorized into three sequences containing 188, 134, and 107 images, respectively. These images contained parts of the Pegasus, Equuleus, and Aquarius constellations. The annotation process for RSOs was manually conducted using Astro ImageJ, a specialized image display environment with tools designed for the calibration and reduction of astronomy-specific images [39]. Using the graphical user interface, the pixel coordinates of RSOs were annotated throughout the sequences. Astrometry.net was used for star annotation. The annotated data served as the ground truth for validating the algorithms’ findings.

To the best of the authors’ knowledge, this dataset is the first of its kind, offering images with a wide field of view that were passively acquired from a near-space environment. Consequently, the identification of resident space objects (RSOs) within the dataset, as well as any metadata that could enable the linkage of detected objects to verifiable resident space objects, will not be disclosed due to the sensitivity associated with them. The primary purpose of the dataset is to facilitate algorithm development. Additional research is currently being conducted to address the sensitivity concerns associated with this dataset.

## 5. Results

### 5.1. Metrics Used

The accuracy of the detection algorithms was gauged using three primary metrics: the precision, recall, and F1 score. The F1 score, which is the harmonic mean of the precision and recall, is a weighted average that favors the lowest-scoring metric. It serves as a general indicator of an algorithm’s performance and is a better-suited metric due to its ability to account for class discrepancies. Additional secondary metrics, including true positives (TPs), false positives (FPs), and false negatives (FNs), were used to compute these primary metrics.

In object detection, TPs denote correctly identified RSOs, FPs represent stars misidentified as RSOs, and FNs are RSOs that the algorithms failed to identify. In streak detection, FPs indicate detected regions that exist outside the annotated values. TPs correspond to annotated values within detected regions. FNs pertain to annotated values that are situated outside the detected regions.

### 5.2. Results of Object Detection

AFD performed optimally when there was camera movement because the displacement of stars between successive frames was typically minimal. This was an advantage over MFD, which subtracted non-consecutive frames that were taken at different times, resulting in a potential misalignment of stars during differencing. Thus, for longer sequences or those containing camera jitter, AFD is the recommended algorithm. The ability to retain the shape of RSOs after differencing gave MFD an edge over AFD. AFD contained overlapping stars and RSOs in adjacent frames before subtraction, resulting in reduced distinctions between static and moving objects in the resulting differenced frame. Because MFD resulted in fewer misidentifications of stars as RSOs, it is the preferred algorithm when higher precision is desired. PFT offered a significant advantage over the other techniques by substituting size filtering with tracking, which was more effective at filtering out small hot pixels that existed in a singular frame. Moreover, because PFT omitted frame differencing techniques, it avoided the visual artifacts that arose from the subtraction processes found in both AFD and MFD. These artifacts in the latter algorithms necessitate additional filtering layers for removal. Hence, PFT is the preferred algorithm for greater recall, as the use of fewer filtering techniques minimizes the loss of true RSOs after preprocessing. The overall performance of these techniques is tabulated in Table 3.

In terms of overall performance, PFT outperformed both AFD and MFD, with an F1 score of 82%, while AFD had the lowest performance at 68%, which was primarily due to its lower precision of 73% compared to MFD and PFT, which were both at 95%. Recall presented challenges for all three algorithms, regardless of their processing techniques. Testing without any filtering during processing yielded maximum recall scores of 71%, 75%, and 87% for the respective sequences. These results highlight that the preprocessing phase significantly impacted the recall, as even without constraints during processing, the recall did not approach 100%. This suggests that preprocessing that involves thresholding and normalization filters out faint RSOs along with noise, resulting in a lower recall.

In the three sequences analyzed, which each presented distinct challenges, the performance of the detection algorithms varied. In the first sequence, which featured a mix of faint and prominent RSOs, MFD excelled in precision but struggled with recall due to its extensive filtering layers. Conversely, in the second sequence, which was primarily composed of faint RSOs, all algorithms faced recall issues because the preprocessing thresholding filtered out many faint RSOs, necessitating a trade-off between recall and filtering. In the third sequence, which was dominated by prominent RSOs, MFD marginally outperformed PFT due to its aggressive filtering, achieving higher precision and a somewhat unexpected higher recall rate, which was possibly attributed to PFT’s tracking inconsistencies with RSOs covering substantial distances between successive frames.

AFD performed the worst in all three sequences, which was primarily due to its poor precision relative to the other algorithms. This was a result of RSOs and stars appearing nearly identical after the subtraction process, which caused difficulties in differentiating between static and moving objects. Additionally, intermittent periods of long frame times resulted in RSOs being detected twice, as objects were too far apart in adjacent frames to combine bounding boxes. This variability in frame time is difficult to account for during processing, as it is a problem inherent to the dataset.

PFT emerged as the best-performing method overall, as it relied on proximity filtering to detect RSOs rather than frame differencing. This proved useful in improving its recall rate, as fewer true RSOs were filtered out during processing. Frame differencing algorithms such as AFD and MFD rely heavily on filtering to account for visual artifacts that remain after subtraction, resulting in their comparably poor recall rates. In addition, rudimentary tracking proved useful for filtering out objects with limited movement, rather than relying on shape, size, and brightness for filtering, as in AFD and MFD; this is a characteristic of unreliable RSO identification.

### 5.3. Results of Streak Detection

In Figure 9, we present the streaks detected in all three image sequences, totaling twenty streaks. These images resulted from stacking the output from each set within its respective sequence. Table 4 presents the mean streak length in pixels and the signal-to-background ratio (SBR) for each sequence. A streak caused by the release of the ballast is visible and highlighted in the green box located on the left side in the first sequence. Although this streak is relatively narrow, it meets the area-based threshold for detection. The precision, recall, F1 score, and accuracy values for the sequences are tabulated in Table 5.

Overall, the algorithm exhibited its poorest performance in the first sequence, where it achieved an F1 score of 65%. This deficiency in performance can be attributed to the low recall, especially in sequences featuring ballast streaks and faint RSOs. It is important to note that a precision of 100% was achieved. This might have been a result of the specific definitions of TPs and FPs, which were customized for streak detection with manual annotations made for object detection. There is a possibility that the algorithm might have overperformed since it was designed to perform optimally with stable data from RSONAR, thus prioritizing the simplification of complex image processing and emphasizing detection accuracy. The relatively low recall values further suggest that the algorithm encountered challenges in identifying RSO streaks that fell below the noise threshold.

## 6. Conclusions

### 6.1. Summary

In this study, we demonstrated three accessible and effective RSO detection methods that used frame differencing and are capable of enhancing space situational awareness. Combining both conventional tracking and proximity filtering techniques proved to be most effective in identifying moving objects under a variety of conditions. We also implemented streak detection using low-resolution wide-field-of-view optical imagery, with a focus on minimal image processing. With the positional data collected from each algorithm, the trajectory of potentially harmful space debris can be predicted and used to mitigate future collisions, thus ensuring that important communication, navigation, and space observation satellites remain unharmed. As the amount of space debris increases at an unprecedented rate, the democratization of detection techniques becomes necessary for protecting the Earth’s orbital environment for decades to come. Table 6 summarizes the optimal detection methods for various applications according to the results described in the previous section.

### 6.2. Future Work

To enhance the performance of the algorithms, several key adjustments should be considered. First, in the case of AFD, it is essential to reduce and stabilize the frame time during processing. This modification would bring RSOs closer together in consecutive frames, thus mitigating double detections and, ultimately, improving the precision of AFD. Additionally, minimizing camera jitter within the dataset would have a positive impact on precision across all three algorithms, particularly when distinguishing stars with motion patterns resembling those of RSOs. An alternative approach involves analyzing object trajectories and separating camera motion from object motion to reduce overlap between stars and RSOs. Furthermore, for both AFD and MFD, replacing size filtering with tracking is recommended, as it can significantly increase the recall rates by allowing faint RSOs to be detected while also reducing the filtering of hot pixels and visual artifacts. Although this may slightly affect the precision, especially in MFD, the impact can be effectively mitigated with a consistent frame time. Finally, it is advisable to avoid overly aggressive thresholding during preprocessing, as it can lead to a significant drop in recall rates by filtering out faint RSOs and background noise. Instead, relying on OpenCV’s contour detection can provide better recall, but it is essential to consider additional post-processing filtering to maintain precision when handling the increased number of detected objects, thereby reducing the risk of misidentifying stars as RSOs.Traditionally, frame differencing techniques were employed during the development of such algorithms, which necessitated the reduction of the image bit depth. However, retaining images at 16 bits could potentially improve the detection metrics, albeit with consideration for the hardware limitations that this may impose.

Currently, streak detection identifies streaklets separately because of the data themselves. The next step is to add margins to the end of each streak to see if streaklet merging occurs in subsequent subsets. Additionally, the RA and Dec values of streaks can be used for correlation and identification with known catalogs, while the streak magnitude can be evaluated using the visual magnitudes of reference stars present in an image.

To further validate these methods, ground-based observation campaigns were conducted using the same imaging system. These observations established the effectiveness of the algorithms with images obtained from the ground, indicating their potential applicability across various observational platforms. Future studies will comprehensively compare ground-based and stratospheric RSO detection while utilizing the same imaging system. Additionally, future research will explore the adaptability and effectiveness of these algorithms under various observational conditions.

## Figures and Tables

**Figure 1 sensors-23-09668-f001:**
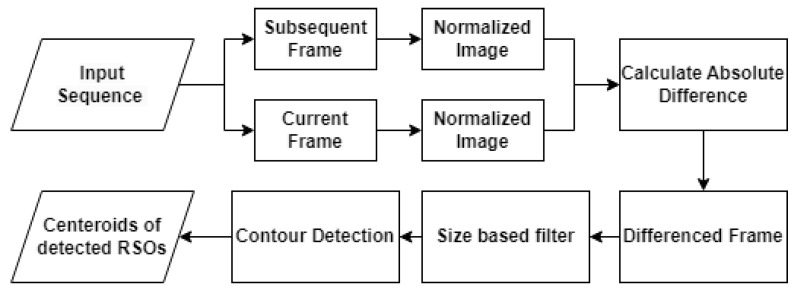
Flow diagram of adjacent frame differencing.

**Figure 2 sensors-23-09668-f002:**

Visualization of AFD processing: (**a**) current frame, (**b**) subsequent frame, and (**c**) differenced frame; the RSOs are highlighted with a green bounding box, and visual artifacts that are to be filtered are highlighted with red boxes.

**Figure 3 sensors-23-09668-f003:**
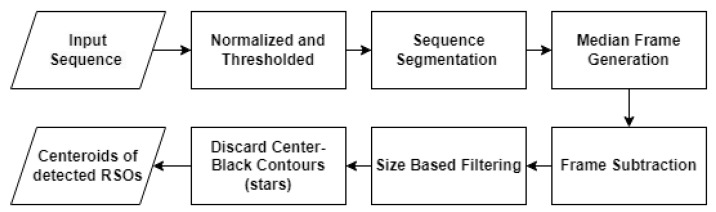
Flow diagram of median frame differencing.

**Figure 4 sensors-23-09668-f004:**
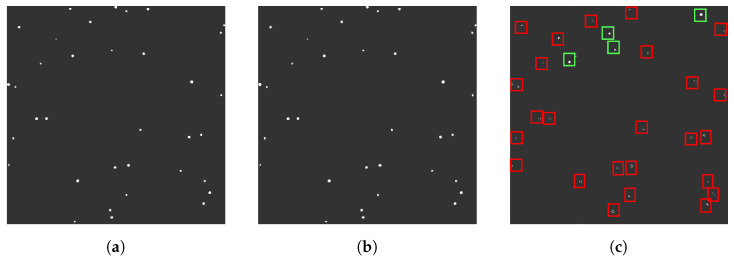
Various frames used in MFD processing: (**a**) the current frame, (**b**) median frame, and (**c**) differenced frame, in which the visual distinction between RSOs and stars can be observed. RSOs maintain their shapes and are highlighted with green bounding boxes, whereas stars and artifacts, which display irregular shapes, are contained within red bounding boxes.

**Figure 5 sensors-23-09668-f005:**
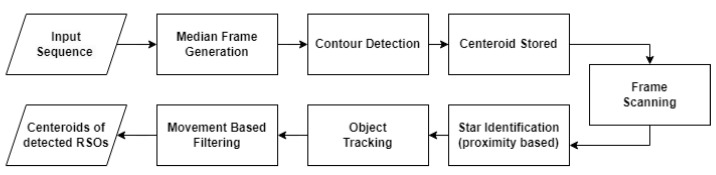
Flow diagram of proximity filtering and tracking.

**Figure 6 sensors-23-09668-f006:**
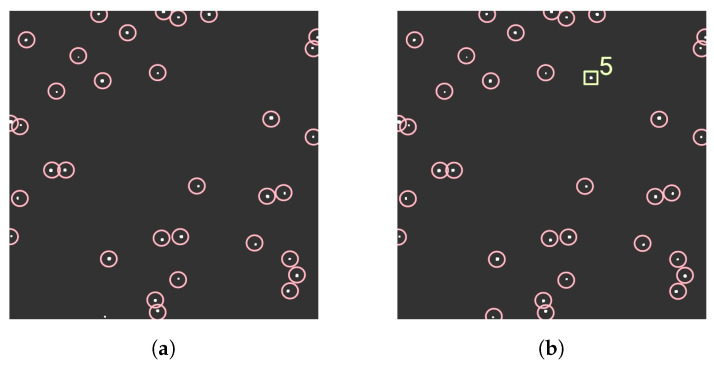
Various frames used in PFT processing: (**a**) the median frame, where all contours present within the images are highlighted with red bounding circles, and (**b**) the current frame, where an RSO is tracked with a unique ID and enclosed in a green bounding box.

**Figure 7 sensors-23-09668-f007:**
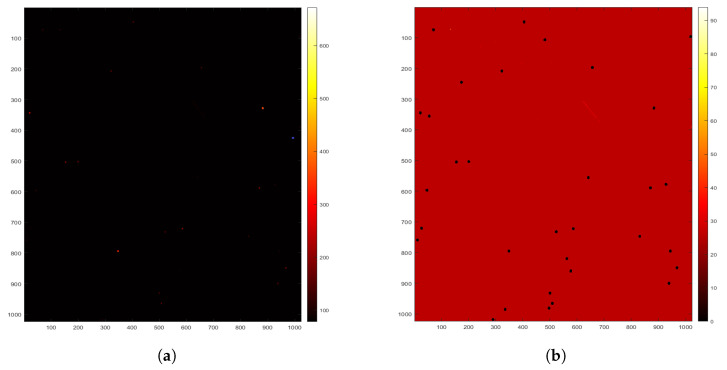
Before star removal and after star removal; the pixel intensity scales are displayed as follows: (**a**) the original image before star removal, where the red dots indicate stars with high pixel values; (**b**) the same image after the stars were removed, with black dots indicating regions where stars were previously present, corresponding to the red dots from (**a**).

**Figure 8 sensors-23-09668-f008:**
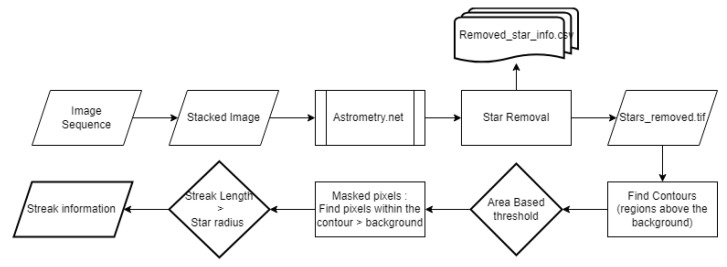
Flow diagram of streak detection.

**Figure 9 sensors-23-09668-f009:**
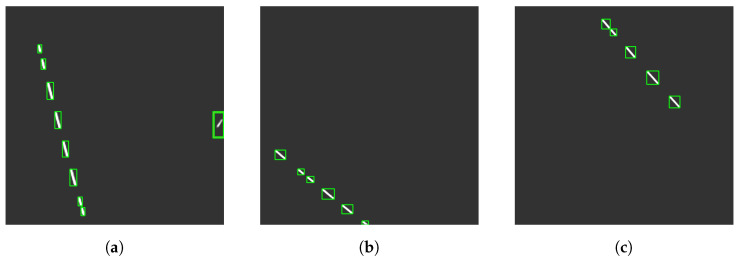
All detected streaks are highlighted with green bounding boxes: (**a**) first sequence, (**b**) second sequence, and (**c**) third sequence. The figures were enhanced using the ZScale algorithm to provide visual clarity.

**Table 1 sensors-23-09668-t001:** Comparison of the detection methods.

Algorithm	Description	Pros	Cons
Frame differencing and background subtraction	Models the background using a running average. The background is subtracted from frames in sequence, and leftover pixels are in motion.	Simple implementation. Computationally inexpensive. Adapts to changing backgrounds.	Does not account for uninteresting motion (i.e., motion due to background objects moving). Limited to a fixed camera; relies on frames aligning with background.
Optical flow	The 2D motion vector for each pixel of an image is computed by comparing it with the next image.	Accurate; measures motion at pixel level.	Requires detailed features for effective use; RSOs may not be detailed enough.
Nearest neighbor	Objects are associated between images by finding the objects closest to them in the next image.	Can perform further analysis to differentiate the movements of RSOs and stars using position/velocity.	Not robust to overlapping RSOs/stars, illumination changes, or fast-moving RSOs.
Streak detection	Models use the plate solver for star removal for streak detection.	Avoids additional blurring. Accounts for uncalibrated images.	Needs Astrometry.net. The observer’s motion can heavily influence the output.

**Table 2 sensors-23-09668-t002:** Imaging parameters used by RSONAR during flight.

Characteristic	Values
Aperture	12.5 mm
Bit depth	16 bits
Chromaticity	Monochrome
Exposure time	100 ms
Field of View	29.7${}^{\circ }$
Focal length	25 mm
Pixel size	6.5 μm / pixel
Pixel scale	104 arcsec/pixel
Quantum efficiency	82%
Resolution	1024${}^{2}$ pixels

**Table 3 sensors-23-09668-t003:** Overall performance metrics for object detection.

Method	Precision	Recall	F1 Score	TP	FP	FN
AFD	73%	63%	68%	387	143	226
MFD	95%	65%	77%	397	22	216
PFT	95%	73%	82%	447	25	166

**Table 4 sensors-23-09668-t004:** Sequence of images and their respective mean streak lengths (in pixels) and signal-to-background ratios (in dB) for the first, second, and third sequences.

Sequence Order	Mean Streak Length (Pixels)	SBR (dB)
First	56.51	27.32
Second	49.8	27.67
Third	61.04	26.64

**Table 5 sensors-23-09668-t005:** Precision, recall, F1 score, and accuracy values for each sequence.

Sequence Order	Precision	Recall	F-1 Score	TP	FP	FN
First	100%	48%	65%	158	0	169
Second	100%	71%	83%	117	0	47
Third	100%	79%	88%	99	0	27

**Table 6 sensors-23-09668-t006:** Summary of the performance of the detection methods with various parameters.

Parameters	Optimal Algorithms
Onboard Processing	MFD
Real-time Processing	PFT
Least False Negatives	Streak Detection
FPGA Implementation	MFD
Changing Light Conditions	Streak Detection
Best Accuracy	Streak Detection
Most Flexible	MFD

## Data Availability

The data presented in this study are available upon request from the corresponding author. Their public availability is currently restricted due to ongoing research and the presence of sensitive information.

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
