# Peer review of "Comparative Analysis of Resident Space Object (RSO) Detection Methods"

_sensors, 2023, doi:10.3390/s23249668_

Round 1

Reviewer 1 Report

Comments and Suggestions for Authors

1. The methods proposed in the paper lack innovation, and the algorithms introduced have practical applications that date back many years. 2. The paper employs a novel dataset; however, due to sensitivity concerns, its usage is restricted. The readers' primary interest lies in this dataset, and the authors should make a thorough comparison between this dataset and other commonly used datasets. Particularly in the experimental section, the same methods should be applied to different datasets to demonstrate whether this dataset possesses unique characteristics. 3. In the experimental section of the paper, when comparing the proposed method with existing methods, it is advisable to use the same dataset or the same sequence of images as the basis for comparison. This approach is essential to demonstrate the distinctions between the methods.

Reviewer 2 Report

Comments and Suggestions for Authors

The "Research Objectives" subparagraph does not seem very clear to me. Both lines 83 and 113 talk about "stare mode". I think the correct term is "sidereal tracking mode", right?!

The second paragraph "Survey of Object..." cites some interesting papers for object recognition using AI, even if  nothing is applied  of what has been said.

It is not clear why talking about images taken from the stratosphere (line 9, line 341 and paragraph 4), but then in line 331 it is stated that a simulated datasets is used. Furthermore the images are taken from the stratosphere it does not mean that the same proposed methods will also work with images acquired from the ground. This is why the conclusions seem a bit "generous" to me.

The images shown are not particularly clear.- The contrast of all images should be increased to make them see better (for example the ZScale algorithm)

It is not clear whether these methods that are proposed are manual, semi-manual or automatic.

- It is not clear which observation mode was used for all the images.

- For the algorithms presented, a flow diagram could be placed to facilitate understanding of the operations.

- For each algorithm presented, the input image should be shown and then the objects found (stars and satellites) clearly with bounding boxes.

- Images are reduced from 16 bits to 8 bits thus reducing the quality of the edges. I don't think this is generally done for this type of algorithm.

- In Figure 2 the object is not highlighted. If the object is not present it would be appropriate to put one with the object to make the algorithm understand.

- In Figure 3 it is not clear which objects are identified. The photo looks identical. It would be better to put bounding boxes on found objects.

Round 2

Reviewer 1 Report

Comments and Suggestions for Authors

1.The article still lacks innovation. Most images in the space debris monitoring field are far more uniform and simplistic compared to images in other areas of machine vision. The primary challenge lies in detecting targets in situations with low signal-to-noise ratios. Clearly, the main technology presented in this paper has not addressed the primary issues within this field, nor proposed innovative technical solutions

2. The main value of the article lies in the utilization of a novel dataset collected by telescopes carried by stratospheric balloons. This approach holds significance within the field of space situational awareness. However, the article fails to conduct an in-depth analysis of this dataset. Additionally, it appears that the author may not be able to continue or complete further extensive work on this aspect within a short period.

Nevertheless, considering the significance of this dataset, I recommend accepting the article for publication. I would also like to politely remind the author to pay attention to copyright, ownership, and usage rights regarding this dataset. After removing sensitive information, it would be beneficial to make this dataset publicly available.